Potential molecular mechanisms of ETV6-RUNX1-positive B progenitor cell cluster in acute lymphoblastic leukemia revealed by single-cell RNA sequencing

Qu Ning 1
Wan Yue 2
Sui Xin 3
Sui Tianyi 4
Yang Yang 5 cmuyy2007@sina.com
1 Pediatrics Department, Jinzhou Central Hospital , Jinzhou , China
2 Oncology Department, Jinzhou Central Hospital , Jinzhou , China
3 Neurosurgery Department, The First Affiliated Hospital of Jinzhou Medical University , Jinzhou , China
4 Clinical Medicine Department, Dalian Medical University , Dalian , China
5 Neurosurgery Department, Jinzhou Central Hospital , Jinzhou , China
Guan Fanglin
Electronic publication date: 2024 Nov 1
Publication date: 2024
Volume: 12
Electronic Location ID: e18445
Received 2024 Aug 2; Accepted 2024 Oct 11
Copyright: © 2024 Qu et al.
Copyright year: 2024
Copyright holder: Qu et al.
License: This is an open access article distributed under the terms of the Creative Commons Attribution License, which permits unrestricted use, distribution, reproduction and adaptation in any medium and for any purpose provided that it is properly attributed. For attribution, the original author(s), title, publication source (PeerJ) and either DOI or URL of the article must be cited.
License URL: https://creativecommons.org/licenses/by/4.0/

Keywords: ETV6-RUNX1, Acute lymphoblastic leukemia, Immune landscape, B progenitor cells, Chromosome amplification, Cell cycle

Funding: The authors received no funding for this work.

==============================
Aim

This study was to explore role of immune landscape and the immune cells in acute lymphoblastic leukemia (ALL) progression.

Background

The most prevalent genetic alteration in childhood ALL is the ETV6-RUNX1 fusion. The increased proliferation of B progenitor cells could expedite the disease’s progression due to irregularities in the cell cycle. Nevertheless, the mechanisms by which particular cell clusters influence the cell cycle and promote the advancement of ALL are still not well understood.

Objective

This study was to explore role of immune landscape and the immune cells in ALL progression.

Methods

Single-cell RNA sequencing (scRNA-seq) data of ETV6-RUNX1 and healthy pediatric samples obtained from GSE132509 were clustered and annotated using the Seurat package, and differentially highly expressed genes identified in each cluster were analyzed using DAVID for pathway annotation. Chromosome amplification and deletion were analyzed using the inferCNV package. SCENIC evaluated the regulation of transcription factors and target gene formation in cells. cellphoneDB and CellChat were served to infer ligand-receptor pairs that mediate interactions between subpopulations. The role of the target gene in regulating ALL progression was assessed using RT-qPCR, Transwell and scratch healing assays.

Results

The bone marrow mononuclear cells (BMMCs) from ETV6-RUNX1 and healthy pediatric samples in GSE132509 were divided into 11 clusters, and B cell cluster 1 was identified as B progenitor cell, which was amplified on chromosome 6p. B progenitor cells were divided into seven clusters. Expression levels of amplified genes in chromosome 6p of B progenitor cell cluster 5 were the highest, and its specific highly expressed genes were annotated to pathways promoting cell cycle progression. Regulons formed in B progenitor cell cluster 5 were all involved in promoting cell cycle progression, so it was regarded as the B progenitor cell cluster that drives cell cycle progression. The key regulator of the B progenitor cell is E2F1, which promotes the migration and invasion ability of the cell line HAP1. The major ligand-receptor pairs that mediate the communication of B progenitor cell cluster 5 with cytotoxic NK/T cells or naive T cells included FAM3C−CLEC2D, CD47−SIRPG, HLAE−KLRC2, and CD47−KLRC2. HLAE−KLRC1 and TGFB1−(TGFBR1+TGFBR2).

Conclusion

This study outlined the immune cell landscape of ETV6-RUNX1 ALL and identified chromosome 6p amplification in B progenitor cells, described the major B progenitor cell cluster driving cell cycle progression and its potential regulatory mechanisms on NK cells and T cells, providing cellular and molecular insights into ETV6-RUNX1 ALL.

Introduction

Acute lymphoblastic leukemia (ALL) is the most common hematological malignancy in children aged 3–6 years and is characterized by abnormal proliferation of immature lymphocytes (Lejman et al., 2022; Atteia, 2023; Nekoeian et al., 2022). Due to many years of research and the utilization of high-throughput sequencing techniques, it is now understood that ALL comprises several subtypes determined by specific somatic genetic abnormalities, some of which affect treatment responses and overall outcomes (Iacobucci & Mullighan, 2017). B-cell acute lymphoblastic leukemia (B-ALL) is a type of blood cancer that involves the uncontrolled growth of B-lymphoid progenitor cells within the bloodstream. Data from the SEER (Surveillance, Epidemiology, and End Results Program) database indicate that the overall survival rate for children with B-ALL over a 5-year period is approximately 89% (Ma, Sun & Sun, 2014). Prognostic indicators for B-ALL encompass a range of factors related to the disease as well as individual patient attributes. Notably, clinical features such as patient age, the count of white blood cells at the time of diagnosis, cytogenetic profiles, and chemotherapy response have been recognized as significant for prognosis in individuals with B-ALL (Brown et al., 2021). In recent studies, there has been a growing body of evidence linking the makeup and dysfunction of immune cells to the effectiveness of clinical treatment and overall prognosis in patients with cancer (Wang & Li, 2019; Savas et al., 2018; Lai et al., 2022).

A typical example of an oncogene produced in utero in B-ALL is ETS translocation variant 6 (ETV6)-runt-related transcription factor 1 (RUNX1) fusion, its presence was only associated with B cell precursor (pB)-ALL. ETV6-RUNX1 pB-ALL is a malignant clonal disorder arising from a solitary cell, distinguished by the buildup of immature B-cells that display characteristics similar to the normal phases of B-cell maturation. Ultimately, this progression results in the inhibition of healthy hematopoiesis and the invasion of several critical organs (Rodriguez-Hernandez et al., 2017). ETV6-RUNX1 accounts for 25% of pediatric B-ALL cases (Wray et al., 2022), and the intratumor tumor microenvironment (TME) and the molecular events involved in this form remain elusive. Numerous recent studies at the single-cell level have described the immune microenvironment associated with human solid tumors, within which malignant transformation is accompanied by significant reorganization of this immune environment (Azizi et al., 2018; Puram et al., 2017). It has been proposed that these alterations may aid in the promotion of tumor growth (Witkowski et al., 2020). Therefore, it is imperative to delineate the immune cell landscape of ETV6-RUNX1 fusion ALL and gain insight into the molecular mechanisms of immune cells that dominate its progression.

The emergence and optimization of single-cell RNA sequencing (scRNA-seq) analysis as a powerful tool to characterize TME has greatly expanded our understanding of the diversity and fate of cells present in the bone marrow (Lee et al., 2021; Shahrajabian & Sun, 2023). A scRNA-seq analysis of T cells from patients with B-ALL patients specifically identified two exhausted T cell populations that exhibited significant heterogeneity in molecular features and broad diversity in clonal derivation (Wang et al., 2021). Zhang et al. (2023) identified 12 heterogeneous B cell populations in scRNA-seq analysis of Ph-like ALL patients with a novel TPR-PDGFRB fusion gene, revealing a comprehensive profile and dynamic changes in cell composition. Sequencing analysis of 1,507 single cells isolated from BM from four pediatric T-ALL patients by De Bie et al. (2018) and coinvestigators determined clonal heterogeneity in primary T-ALL samples and identified genomic lesions that initiate T-ALL in multipotent progenitor cells. Although scRNA-seq analysis has led to significant advances in our understanding of cellular heterogeneity in ALL, studies that specifically applied this technology to ETV6-RUNX1 fusion ALL to provide insights are rare.

In this study, we specifically selected the bone marrow mononuclear cells (BMMCs) scRNA-seq data of ETV6-RUNX1 and healthy pediatric samples from GSE132509 data sets for analysis, not only to explore the landscape of TME, but also to explore the immune cells that dominate its progress and the potential molecular mechanisms involved.

Materials and Methods

Acquisition of ALL sequencing data at the cellular level, capture and clustering of high-quality cells

Single cell gene expression data of MMCs from four ETV6-RUNX1 and three healthy pediatric samples were downloaded from GEO database with ID GSE132509, and import them into Seurat (Butler et al., 2018) package to read with Read10X function, imported into the Seurat package for reading using the Read10X function. Quality control was performed with 200–5,000 genes and the proportion of mitochondrial genes <10% as the threshold. SCTransform function was created to conduct normalization, variance stabilization, and feature selection based on a unique molecular identifiers (UMI)-based gene expression matrix. Then PCA was performed on all genes and applied the first 20 dimensions to compute a UMAP embedding. After clustering the cells, each subgroup was annotated according to the marker genes provided by CellMarker (Hu et al., 2023), Cell Taxonomy (Jiang et al., 2023) and PanglaoDB (Franzen, Gan & Bjorkegren, 2019) databases.

Differential expression analysis

The differential expression analysis between cell clusters was performed by the FindAllMarkers function, and the parameters were set to logfc.threshold=0.25, min.pct=0.25, only.pos=TRUE to screen differentially expressed genes (DEGs). DEGs between ETV6-RUNX1 and healthy pediatrics samples for B progenitor cell were screened by the difference analysis of FindMarkers function with the threshold of |avg_log2FC| > 0.25 and p_val_adj < 0.010.01.

Pathway annotation analysis

The required target gene set was imported into the DAVID database (Dennis et al., 2003), selected the species as “homo sapiens” and the type as “gene list”, and then submited the List to obtain the chart of enriched pathways. The pathways with p < 0.05 were selected and called “ggplot” to visualize the results as a bar graph with the count value. KEGG pathway annotation analysis was performed using the gseKEGG function of clusterProfiler package (Wu et al., 2021), and the results were visualized using the GseaVis package.

Copy number variation analysis

The inferCNV package (Patel et al., 2014) was served to analyze copy number variations (CNVs) in chromosomal regions of progenitor B cells in ETV6-RUNX1 samples. Raw count matrices, annotation files, and gene/chromosome location files were prepared, and cytotoxic NK/T cells in healthy pediatric samples were set as reference cells, and evaluated CNV with the parameters of cluster_by_groups=TRUE, analysis_mode=“subclusters”, HMM_type=“i3”, denoise=TRUE, HMM_report_by=“subcluster”, HMM=TRUE.

Single-cell regulatory network inference and clustering analysis

Single-cell regulatory network inference and clustering (SCENIC) is a method to identify stable cell states by using single-cell RNA-seq data to evaluate the activity of gene regulatory networks (GRN) in each cell (Aibar et al., 2017). Here, the expression matrix of transcription factors (TFs) was input into GENIE3 of SCENIC to form the co-expressed gene set using top5perTarget. Motif enrichment analysis was performed for each co-expression module using RcisTarget, and significantly enriched motifs were retained for TF annotation and gene scoring in the co-expression module. Genes with low motif scores in the co-expression module were deleted, and the remaining output was regulon file. A regulon is the gene set of a TF and its corresponding target gene. Finally, cytoscape was used to visualize regulon.

Cell communication analysis

CellphoneDB (Efremova et al., 2020) was used to construct a cell subset ligand-receptor interaction network, and ktplots package was employed to visualize ligand-receptor pairs with inhibitory effects on T cell activity as bubble plots. The CellChat package (Jin et al., 2021) was served to calculate the number of ligand-receptor pairs between cell subsets, and netVisual_bubble function was ed to display the bubble map of the degree of interaction of ligand-receptor pairs related to TGF-β signaling pathway.

Cell culture and transfection

Human bone marrow stromal cells HS-5 (BNCC339313) were purchased from BNCC (Beijing) Biotechnology Co. Ltd. and the chronic granulocytic leukemia cell line HAP1 (item #C631 bath 29663) was purchased from Horizon Discovery. Cells were cultured in Dulbecco’s modified Eagle medium (Gibco, Waltham, MA, USA, 11965) and supplemented with 10% fetal bovine serum (Gibco, Waltham, MA, USA, 26140-095) and 1% antibiotics (Gibco, Waltham, MA, USA, 15070-063). -092) cultured in Dulbecco’s Modified Eagle medium (Gibco, Waltham, MA, USA, 11965-092) supplemented with 10% fetal bovine serum (Gibco, Waltham, MA, USA, 26140-095) and 1% antibiotics (Gibco, Waltham, MA, USA, 15070-063). Cells were cultured at 37 °C and 5% CO2. Lipofectamine 2000 (Invitrogen, Carlsbad, CA, USA) was used to transfect cells with negative control (NC) and E2F1 siRNA (Sagon, Yangshan, China). The target sequence of E2F1 siRNA was: sense, 5′-GACGUGUCAGGACCUUCGU-3′, antisense, 5′-ACGAAGGUCCUGACACGUC-3′.

QRT-PCR

Following the extraction with TriZol (Thermo Fisher, Waltham, MA, USA), the RNA was quantified in a spectrophotometer (Thermo Fisher, Waltham, MA, USA) and reverse transcribed to cDNA by Qiagen One-Step RT-PCR kit (Qiagen Gmbh, Hilden, Germany). One μg cDNA was then subjected to qRT-PCR experiments. Amplification experiments were performed in an ABI 7500 system (Thermo Fisher, Waltham, MA, USA) using SYBR Green. The 2−ΔΔCT method was applied for the calculation on the mRNA level with GAPDH as the reference gene (Livak & Schmittgen, 2001).

Scratch healing test

The cell lines were inoculated in 6-well plates, and when the adherent wall grew throughout the bottom, a scratch was made vertically by using a 200 μL pipette tip, rinsed twice with phosphate buffer, and photographed under an inverted microscope after 0 and 48 h of the scratch, respectively. Scratch healing rate = (0 h width − 48 h width)/0 h width × 100%, test three times, re-well two times.

Transwell analysis

A total of 30 μL of Matrigel was applied to the bottom of the upper chamber (serum-free medium) of the Transwell experimental setup (Corning, New York, NY, USA), followed by inoculation of cells into the upper chamber, followed by addition of PRMI-1640 medium containing 10% FBS to the lower chamber, and after incubation at 37 °C for 1 d, the non-migrated cells in the upper chamber were removed. Migrated cells were fixed with 4% paraformaldehyde and stained with crystal violet for 30 min. Cells were observed under a microscope and counted.

Statistical analysis

All statistics were analyzed using R software. The rank sum test was used to compare the significance of differences between two groups of continuous variables, and the significance of differences among three or more groups of continuous variables was calculated by the kruskal-wallis test. A significant difference was set at P < 0.05.

Results

ScRNA-seq depicted the landscape of BMMCs from ALL patients with ETV6-RUNX1

To characterize the landscape of BMMCs from ALL patients with ETV6-RUNX1, we performed analysis of scRNA-seq data and separated BMMCs into 11 clusters, involving cell types including in cell types including B cell, erythrocytes, monocytes, cytotoxic NK/T cells, naive T cells. The B cells distributed in five separate clusters, and erythrocytes distributed in three separate clusters (Fig. 1A). As shown in Figs. 1B, 1C, based on the expression of markers CD24, BANK1 and CD19, we found that B cell cluster 1 and B cell cluster 2 had similar marker expression profiles. B cell cluster 3 and B cell cluster 5 had similar marker expression profiles. The most obvious expression marker of B cell cluster 4 compared with other four B cell clusters was BANK1. CD3D is a specific marker gene for T cells and is therefore highly expressed in both cytotoxic NK/T cells and naive T cells. CCL5 is expressed in cytotoxic NK and T cells thereby promoting their migration (Li et al., 2020). NKG7 is also specifically expressed by CD8 T cells and NK cells (Li et al., 2022). What distinguishes cytotoxic NK/T cells from naive T cells was CCR7, a chemokine receptor that controls homing to secondary lymphoid organs and is specifically expressed in naive T cells of T cells (Lewis, Tarlton & Cose, 2008). The chemotactic cytokine CCL5 is extensively overexpressed in malignant lesions to guide T cell infiltration (Huffman et al., 2020). HBA1 and HBB were expressed in mature erythrocyte as α-globin and β-globin, respectively (Kumar et al., 2022). High expression of HBA1 and HBB was detected in all three isolated erythrocytes (Figs. 1B, 1C). Significant differences in the proportions of B cells, naive T cells, monocytes, and cytotoxic NK/T cells were detected between ETV6-RUNX1 positive ALL patients and healthy pediatric samples. B cell cluster 1 accounted for the highest proportion of ETV6-RUNX1 positive ALL patients, and was the only one among the four differential clusters that was significantly higher in ETV6-RUNX1 positive ALL patients than in healthy pediatric samples (Fig. 1D), suggesting that B cell cluster 1 may be the major cluster contributing to ETV6-RUNX1 ALL occurrence.

Figure 1 ScRNA-seq depicted the landscape of BMMCs from ALL patients with ETV6-RUNX1.

(A) The UMAP plot shows the cell types corresponding to the cell populations in ETV6-RUNX1 positive ALL patients and healthy pediatric samples. (B) Expression of cell type-specific marker genes in each cluster. (C) Two-dimensional UMAP plot of the distribution of cell type-specific marker genes in each cluster. (D) Differences in the distribution proportions of each cluster between ETV6-RUNX1 positive ALL and healthy pediatric samples. ns: p > 0.05, *p < 0.05.

Inference of the identity of each B cell subgroup

Since B cells is separated into five clusters, the identity of each cluster has yet to be confirmed. The identity of each cluster was confirmed functionally by analyzing the pathway enrichment of significantly overexpressed genes in each cluster compared to other clusters. The top five pathways significantly enriched for the overexpressed genes of B cell cluster 1 were cell division, negative regulation of apoptotic process, mRNA splicing via spliceosome and cell cycle and mitotic cell cycle. Therefore, B cell cluster 1 is a cell cycle-related B cell cluster (Fig. 2A). The overexpressed genes of B cell cluster 2 were significantly annotated to aerobic respiration, oxygen transport and carbon dioxide transport pathways (Fig. 2B). Thus, B cell cluster 2 is the respiration-associated B cell. The overexpressed genes of B cell cluster 3 were significantly annotated in translation, cytoplasmic translation, ribosomal small subunit biogenesis and rRNA processing (Fig. 2C). Therefore, B cell cluster 3 is a B cell cluster related to protein synthesis. Almost all of the pathways of enrichment of specific high expression genes in B cell cluster 4 are related to immunity, including innate immune response, B cell receptor signaling pathway, immune response, positive regulation of B cell activation, antibacterial humoral response, immunoglobulin mediated immune response and humoral immune response (Fig. 2D). The highly expressed genes in B cell cluster 5 were significantly enriched in protein folding, innate immune response, protein targeting to ER, response to virus, positive regulation of T cell differentiation, cellular respiration and activation of innate immune response (Fig. 2E). Because the enriched pathways of B cell cluster 4 and cluster 5 were similar, we analyzed the expression levels of more B cell markers in these two clusters. As shown in Fig. 2F, we found that both B cells 4 and B cells 5 highly expressed marker genes related to plasma B cells, including IGHM, JCHAIN, IGKC, and IGHD (Ferjeni et al., 2022; Li et al., 2021; Schmidt et al., 2021), which suggests that these two B cell types may be plasma B cells, which in turn play a role in secreting antibodies or resisting viruses. Furthermore, we analyzed the highly expressed genes in B cell cluster 1 compared with four other B cell clusters, searched the Cell taxonomy database and compared with the B cell marker genes organized in the CellMarker database. It was found that almost all of these genes are markers of progenitor B cells, and MAD2L1 and H2AFX in these genes are associated with B-lineage acute lymphoblastic leukemia (Gupta et al., 2023). We therefore inferred that B cell cluster 1 is a progenitor B cell (Fig. 2G).

Figure 2 Inference of the identity of each B cell subgroup.

(A–E) Pathways significantly annotated by the respective overexpressed genes in B cell cluster 1, cluster 2, cluster 3, cluster 4, and cluster 5. (F) Expression of B-cell markers, including IGHM, JCHAIN, IGKC and IGHD in five B-cell clusters. (G) Expression of B progenitor cells markers in each B cell cluster.

Chromosome 6p amplification and affected genes in progenitor B cells from ETV6-RUNX1 positive ALL patients

Aberrations in 22 pairs of autosomes were explored by creating InferCNV clustered heatmaps for B progenitor cells from ETV6-RUNX1 positive ALL patients. Heatmap showed CNV amplification on chromosome 6p in progenitor B cells from ETV6-RUNX1 positive ALL patients compared to reference cells (Fig. 3A). By performing differential analysis of progenitor B cell marker genes between ETV6-RUNX1 positive ALL samples and healthy pediatric samples, 465 up-regulated DEGs were identified and overlapped with genes that underwent CNV amplification within chromosome 6p, and 38 genes were found in the intersection part (Fig. 3B). We found that these 38 genes were predominantly enriched in pathways associated with antigen processing and presentation, T cell activation and adaptive immunity (Fig. 3C). In particular, the increased expression of these 38 genes may be influenced by CNV amplification, and their expression in ETV6-RUNX1 positive ALL samples and healthy pediatric samples was shown in Fig. 3D. These results imply that the amplification of chromosome 6p may modulate the immune escape mechanism of ETV6-RUNX1-positive ALL cells by up-regulating genes associated with antigen presentation and T-cell activation, thereby altering their immune microenvironment and promoting disease progression.

Figure 3 Chromosome 6p amplification and affected genes in progenitor B cells from ETV6-RUNX1 positive ALL patients.

(A) The InferCNV clustered heatmap of ancestor B cell of ETV6-RUNX1 positive ALL patients, and the reference cell is cytotoxic NK/T cells. Red and blue represent the level of gene expression in the segment chromosomes. The darker the red is, the higher the degree of chromosome amplification is; the darker the blue is, the higher the degree of chromosome deletion is. (B) Overlap analysis of B progenitor cell markers between ETV6-RUNX1 positive ALL patient and healthy pediatric samples and genes that undergo CNV amplification within chromosome 6p. (C) The obtained enriched pathways were analyzed based on the annotation of the expression of 38 CNV amplified genes. (D) Expression profile of 38 genes in ETV6-RUNX1 positive ALL patients and healthy pediatric samples.

Key B progenitor cell clusters tend to drive cell cycle progression

Because progenitor B cells make up the highest proportion of ETV6-RUNX1 positive ALL patients and may be the main cells mediating the progression of ETV6-RUNX1 ALL, progenitor B cells were refined. The B progenitor cells were divided into seven clusters by cluster analysis (Fig. 4A). The amplified genes within chr6p were collected to calculate enrichment scores in each B progenitor cell cluster, and the results showed that the expression level of the amplified genes within chr6p was the highest in B progenitor cell cluster 5 (Fig. 4B). FindAllMarkers analysis identified differentially up-regulated genes for B progenitor cell cluster 5 that were significantly higher than those of the other six B progenitor cell clusters (Fig. 4C), and these genes were annotated to cell cycle-related pathways, including chromosome segregation, mitotic nuclear division, sister chromatid segregation, DNA replication, spindle organization, cell cycle checkpoint signaling (Fig. 4E). Moreover, the enrichment score of cell cycle pathways in B progenitor cell cluster 5 was significantly higher than that in other B progenitor cell clusters (Fig. 4D). Therefore, B progenitor cell cluster 5 was identified as cell cycle related B progenitor cell.

Figure 4 Key B progenitor cell clusters tend to drive cell cycle progression.

(A) Two-dimensional UMAP diagram of B progenitor cell classification. (B) Scores of amplified genes within chromosome 6p in each B progenitor cell cluster calculated by AU Cell. (C) Differentially up-regulated genes in B progenitor cell cluster were significantly higher than the other six B progenitor cell clusters. (D) GSEA shows the enrichment scores of cell cycle pathways in B progenitor cell cluster 5 and other B progenitor cell clusters. (E) Radar plot shows biological processes annotated by significantly highly expressed genes of B progenitor cell cluster 5.

Transcriptional regulatory networks within key B progenitor cell cluster

B progenitor cell cluster 5 has been identified as the key carcinogenic B progenitor cell cluster in ETV6-RUNX1 positive ALL patients. The regulon of the cluster was identified by SCENIC, showing 19 significantly activated TFs, including ATF4 and E2F1, among others (Fig. 5A). Among them, E2F1 regulon and EZH2 regulon are involved in promoting cell cycle progression, these regulons may be the main regulatory signal for B progenitor cell cluster 5 to promote cell cycle progression (Figs. 5B, 5C).

Figure 5 Transcriptional regulatory networks within key B progenitor cell cluster.

(A) AUCell scores of regulons in B progenitor cell cluster 5. (B) Regulons of E2F1 in B progenitor cell cluster 5. (C) Regulons of EZH2 in B progenitor cell cluster 5.

Transcription factors in B progenitor cells regulate leukemia cells affecting ALL

In this study, we explored the relative expression levels of key transcription factors in leukemia cells by RT-qPCR, and the results showed that E2F1 and EZH2 were significantly upregulated in leukemia cell lines compared with normal bone marrow stromal cells (Figs. 6A, 6B). Scratch healing assay demonstrated that the silencing of E2F1 resulted in the down-regulation of migratory ability of leukemia cell lines, implying the promoting effect of E2F1 on the migratory ability of leukemia cell lines (Figs. 6C, 6D). In contrast, Transwell experiments revealed a tangible facilitating effect of E2F1 on the invasion ability of leukemia cell lines (Figs. 6E, 6F). The above findings all illustrate the tangible regulatory role of E2F1, a transcription factor of B progenitor cell cluster, in leukemia progression.

Figure 6 Transcription factor-regulated leukemia cell lines of the B progenitor cell cluster.

(A) mRNA expression levels of E2F1 in HAP1 compared to HS-5. (B) mRNA expression levels of EZH2 in HAP1 compared to HS-5. (C) Relative scratch healing rate of E2F1-silenced leukemia cell lines. (D) Scratch healing results of E2F1-silenced leukemia cell lines at 0 h and 48 h. (E) Quantification of invasion level of E2F1-silenced leukemia cells in Transwell assay. (F) Transwell assay results of E2F1-silenced leukemia cell lines. **p < 0.01, ***p < 0.001, ****p < 0.0001.

Potential regulatory mechanism of key B progenitor cell cluster on NK cells or T cell

As we found that the high expression gene of B progenitor cell cluster 5 was annotated in the process of innate immune response and positive regulation of T cell differentiation in the results of Fig. 2E, we analyzed the communication of B progenitor cell clusters with cytotoxic NK/T cells and naive T cells, respectively. According to the results CellphoneDB analysis, there were 44 pairs of ligand-receptor pairs mediating the communication between B progenitor cell cluster 5 and cytotoxic NK/T cell, and 28 pairs of ligand receptor pairs mediating the communication between B progenitor cell cluster 5 and naive T cell (Fig. 7A). The main ligand-receptor pairs that dominate the communication between B progenitor cell cluster 5 and cytotoxic NK/T cell or with naive T cell included FAM3C-CLEC2D, CD47-SIRPG, HLAE-KLRC2 and HLAE-KLRC1 (Fig. 7B). The same analysis was performed using CellChat, the results showed that the number of ligand-receptor pairs mediating the interaction of B progenitor cell cluster 5 on cytotoxic NK/T cells and naive T cells was 79 and 56, respectively (Fig. 7C). The ligand receptor pairs with the greatest contribution were TGFB1−(TGFBR1+TGFBR2) (Fig. 7D).

Figure 7 Potential regulatory mechanism of key B progenitor cell cluster on NK cells or T cell.

(A) Number of ligand-receptor pairs mediating B progenitor cell cluster 5 communication with cytotoxic NK/T cells analyzed by CellphoneDB. (B) The main ligand-receptor pair that dominates the communication between B progenitor cell cluster 5 and cytotoxic NK/T cell or naive T cell analyzed by CellphoneDB. (C) Number of ligand-receptor pairs mediating B progenitor cell cluster 5 communication with cytotoxic NK/T cells calculated by CellChat. (D) Ligand-receptor pairs involved in B progenitor cell cluster 5 acting on cytotoxic NK/T cells and naive T cells calculated by CellChat.

Discussion

The ETV6-RUNX1 fusion itself arises from a fetal hematopoietic stem cell or a very early B progenitor cell (Bateman et al., 2010). ETV6-RUNX1 can create an aberrant progenitor lineage prone to malignant transformation by accumulating additional minor genes that act as drivers of leukemogenesis (Rodriguez-Hernandez et al., 2017). In this study, we identified B cell, erythrocytes, monocytes, cytotoxic NK/T cells and naive T cells in the BMMC of ETV6RUNX1 positive ALL patients. Among them, B cells were divided into five clusters, and B cell cluster 1 was identified as B progenitor cell according to the specifically expressed gene, which was the highest proportion of all cellular cluster in ETV6-RUNX1 positive ALL patients, and was the only cell cluster with a significantly higher proportion in ETV6-RUNX1 positive ALL patients than in healthy pediatric samples. In addition, this study reveals for the first time the chromosomal expansion of B progenitor cell clusters in ETV6-RUNX1 ALL and its role in cell cycle regulation, clarifying the potential contribution that genes in this expanded region may have in immune escape and immune microenvironment regulation.

ETV6-RUNX1 fusion is not sufficient for the occurrence of ALL. Other genetic alterations involved in cell cycle and B-cell lineage differentiation are required for the transformation of ALL (Greaves, 2018). It is worth mentioning that the preleukemic initiation function of the ETV6-RUNX1 fusion is associated with clonal expansion early in the fetal B-cell lineage (Alpar et al., 2015). One case report found intrachromosomal amplification of chromosome 21 by ETV21/RUNX21 FISH screening for a B-cell precursor ALL (Garcia et al., 2013). In this study, we found that chromosome 6p was amplified in B progenitor cells of ETV6-RUNX1 positive ALL patient, which may be a non-negligible factor contributing to the development of ETV6-RUNX1 ALL. We divided B progenitor cell into seven clusters and identified B progenitor cell cluster 5 as the key B progenitor cell cluster because amplified genes within chromosome 6p were expressed at the highest level in B progenitor cell cluster 5, and multiple regulons involved in cell cycle regulation in B progenitor cell cluster 5 were significantly activated, and the significantly overexpressed genes were annotated to cell cycle-related pathways, which tended to drive cell cycle progression and accelerate the malignant transformation of the disease.

We also found the potential regulatory mechanism of B progenitor cell cluster 5 on NK cells or T cell by cellular communication analysis. The main ligand-receptor pairs that dominate B progenitor cell cluster 5 communication with cytotoxic NK/T cells or naive T cells included FAM3C−CLEC2D, CD47−SIRPG, HLAE−KLRC2, and CD47−KLRC2. HLAE−KLRC1B and TGFB1−(TGFBR1+TGFBR2). It has been reported that the interaction between CD47 and SIRPγ promotes antigen-specific T-cell proliferation and costimulation (Dehmani et al., 2021). The interaction between HLAE and KLRC1 impairs NK cell antitumor activity in solid malignancies, KLRC1 knockout is an effective strategy to improve NK cell antitumor activity against HLA-E tumors (Mac Donald et al., 2023). Activated CD4 T cells can express NKG2C, an activated receptor for HLA-E interaction (Zaguia et al., 2013). Strong HLA-E:NKG2C ligation can contribute to recognition and killing of glioblastoma cells by NK cells (Murad et al., 2022). B cells have a high-affinity TGF-β receptor and secrete TGF-β, which inhibits the proliferation and function of cytotoxic T lymphocytes and NK cells by binding to TβRII to TβRI (Yang, Pang & Moses, 2010). These reports are sufficient to support our findings and allow us to fully appreciate the role of cell cycle-associated B progenitor cell cluster 5 in regulating the cellular activity of NK cells and T cells.

Nonetheless, this study has certain research limitations. Initially, the sample size was limited, and future research will aim to expand it by incorporating patients of various ages and clinical traits, thereby enhancing the generalizability of the findings and minimizing possible biases. In addition, we lack the support of in vivo models (e.g., animal experiments), limiting the physiological relevance and clinical translational potential of the results. Finally, this study is mainly based on single-cell RNA sequencing technology, and in the future, we will integrate multi-omics data and combine proteomics, epigenetics and metabolomics in order to construct a more comprehensive molecular regulatory network.

Conclusion

In this study, we systematically mapped the immune cell landscape of ETV6-RUNX1 ALL by single-cell RNA sequencing, revealing the critical role of B progenitor cell clusters in chromosome 6p amplification. These genes in the expansion region were mainly enriched in pathways related to cell cycle regulation and immune responses, especially antigen presentation, T cell activation and immune escape mechanisms. In particular, B progenitor cell clusters, through ligand-receptor interactions with NK/T cells, may alter the immune microenvironment of ALL and promote the survival and spread of leukemia cells. In particular, the B progenitor clusters mediate the regulation of NK and T cell activity by transduction through FAM3C-CLEC2D, CD47-SIRPG, HLAE-KLRC2, HLAE-KLRC1, and TGFB1−(TGFBR1+TGFBR2). In conclusion, our study provides new perspectives on the development of ALL and potential molecular targets for future targeted therapies.

Supplemental Information

Supplemental Information 1 MIQE_checklist.

Supplemental Information 2 Response_to_editor.

Abbreviations

ALL Acute lymphoblastic leukemia

scRNA-seq Single-cell RNA sequencing

BMMC Bone marrow mononuclear cell

ETV6 ETS translocation variant 6

RUNX1 Runt-related transcription factor 1

TME Tumor microenvironment

TRM Tissue-resident memory T cell

UM Unique molecular identifiers

DEG Differentially expressed gene

CNV Copy number variation

SCENIC Single-cell regulatory network inference and clustering

UMAP Uniform Manifold Approximation and Projection for Dimension Reduction

Additional Information and Declarations

Competing Interests

Author Contributions

Data Availability

The authors declare that they have no competing interests.

Ning Qu conceived and designed the experiments, performed the experiments, analyzed the data, prepared figures and/or tables, authored or reviewed drafts of the article, and approved the final draft.

Yue Wan conceived and designed the experiments, analyzed the data, authored or reviewed drafts of the article, and approved the final draft.

Xin Sui conceived and designed the experiments, analyzed the data, prepared figures and/or tables, and approved the final draft.

Tianyi Sui performed the experiments, authored or reviewed drafts of the article, and approved the final draft.

Yang Yang performed the experiments, analyzed the data, prepared figures and/or tables, and approved the final draft.

The following information was supplied regarding data availability:

Data is available at GEO: GSE132509.

The raw data is available in GitHub and Zenodo:

https://github.com/ningQu197/Raw-data.git

ningQu197. (2024). ningQu197/Raw-data: Raw data (v.1.1.0). Zenodo. https://doi.org/10.5281/zenodo.13120220.

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
