# Peer review of "Potential molecular mechanisms of ETV6-RUNX1-positive B progenitor cell cluster in acute lymphoblastic leukemia revealed by single-cell RNA sequencing"

_PeerJ, doi:10.7717/peerj.18445_

## Round 0.1 · original submission · Major Revisions

We have now received comments from two reviewers, and based on their feedback, I have decided that your manuscript requires major revisions before it can be considered for publication. While one reviewer suggested minor revisions, the other recommended major changes. After careful consideration of both reviews, I believe that addressing the more substantial concerns raised will significantly strengthen your paper. Please carefully consider all the reviewers' comments and suggestions, paying particular attention to the major issues highlighted. We look forward to receiving your revised manuscript that thoroughly addresses these concerns.

Reviewer 1 ·

Basic reporting

no comment

Experimental design

no comment

Validity of the findings

no comment

Additional comments

In this study, the author clarified the specific B progenitor cell cluster affected the B-cell acute lymphoblastic leukemia progression and its underlying molecular mechanisms in this disease. The experimental design and analysis are rigorous and smooth logic, the results and conclusions support each other. However, there are still some deficiencies in details in the manuscript. Please revise it carefully according to the comments below.
1. In the title, the “underlying” is adjective, would it be more appropriate to add an "its" before this word.
2. In the background and results of abstract, the cell cluster with cell cycle-regulating trait was identified, but the important role of cell cycle affected the B-cell acute lymphoblastic leukemia (ALL) progression should be stressed. But the results are seem not strongly related to context and purpose.
3. In introduction, the clinical features, therapies and prognostic survival rate, risk factors, and the current treatment challenges of B-ALL is what. In line 74, the tumor microenvironment (TME) play a crucial role affecting ALL progression, what are the existing studies on TME in ALL, the characteristics of TME is what.
4. Line 71, what is the specific mechanism of the accumulation of B cells inhibiting the function of hematopoietic stem cells. Line 73, “TV6-RUNX1” is incomplete, please complete it. Line 77, scRNA-seq was showed in text at first time, please add its full name.
5. Line 153, the “CO2” is “CO2”. Figure 1A (Line 188-189), the CD24, BANK1 and CD19 were expressed in B cells 1 and B cells 2. How do you distinguish between these two cell clusters.
6. Line 191-197, the marker of each cell cluster can be briefly described, and their functions can be added to the discussion section. Such as CCL5 was highly expressed in NK and T cells.
7. Line 228-232, Please simplify the sentences expression, such as (Line 228-231) the IGKC is a marker of antibody-secreting cells representing the plasma B cells, so the B cell cluster 4 with highest IGKC was presumed as plasma B cells. In addition, Line 227, the B cell cluster 4 was presumed to be naive B cells or follicular B cells, but in Line 231, the B cell cluster 4 is presumed to be plasma B cells. What is the relationship between these cell types, which could be the final definition of B cell cluster 4.
8. Line 249-251, Could the author adjust the description order of Figure 3D and Figure 3C, these genes were associated with the adaptive immune response, antigen processing and presentation.. (Figure 3C), the CNV amplification further enhanced their function (Figure 3D). Line 251-253, Please rewrite the assumptions in this paragraph.
9. In Line 242-243, the CNV amplification on chromosome 6p was found, Could the authors discuss it, which amplification occurs on chromosome 6 and affects the progression of B-ALL upon previous reported studies.
10. What are the limitations of this article, which experiments or methods will be helpful for future research.

Reviewer 2 ·

Basic reporting

The aim of this study was to unravel the B progenitor cell subset driving the cell cycle and its molecular mechanisms in ETV6-RUNX1-positive B-cell acute lymphoblastic leukemia (ALL). The idea of this study is overall conventional. To begin with, the study first processed ALL-related data obtained from public databases by single-cell clustering and revealed the basic profile of chromosome-associated mutations in ALL samples by mutation analysis. The follow-up study mainly focused on the impact of the interaction relationship between cell subpopulations on ALL progression. The study, in addition to bioinformatics analyses, also included a number of cellular experiments, which were mainly used to validate the effects of ALL progression-associated target genes on the migratory and invasive abilities of the cell line HAP1. In conclusion, this study is a relatively comprehensive study, but the following questions still need to be addressed before publication:
1. What is the current status of the ligand-receptor pairs revealed in this study in terms of existing research, and has the mechanism of their regulatory effects on ALL been revealed? If the mechanism has been systematically elucidated in relevant studies, then the discussion section should focus more on the innovative points of these ligand-receptor pairs in this study.
2. Why do B cell subsets interact with NK/T cells? And what does the cell-to-cell interaction suggest about the process of material exchange between the two? What does this tell us about the progression of ALL? All these points are suggested to be elucidated in depth in the introduction.
3. In the elaboration of lines 86-90, it is illustrated through the literature that T-cells in B-ALL patients undergo corresponding dynamic changes, but why does this study focus on T-cells, since it is exploring B-cell related diseases? Please check the literature for accuracy and expand on this study.
4. The paragraphs in the introductory section are not sufficiently distinct, and it is suggested that the first paragraph be split into two paragraphs, one focusing on advances in the study of ALL, the advantages and disadvantages of conventional research tools and treatments, and then a separate paragraph focusing on current advances related to the study of ALL through single-cell analysis.

Experimental design

5. The B progenitor cell-associated marker gene is annotated as being associated with cell cycle regulation in this study, but the mechanism by which this gene regulates the cell cycle needs to be added to the literature to clarify it, and in the meantime, does this gene directly regulate the cycle of B cells or other cells? Please provide additional information on this.

Validity of the findings

6. It is suggested to streamline the description of the results regarding Figure 1 by simply focusing on highlighting what marker genes are differentially expressed between cell subpopulations, without focusing too much on commonalities between cell subpopulations. Suggested changes to this.
7. A focused description of the results of Figure 3 is recommended because the findings in this section reveal genes associated with ALL progression and immunomodulation, and thus it is recommended to elucidate how these genes inform subsequent analyses and to propose reasonable hypotheses as to what the link is between ALL immunomodulation and CNV.

Additional comments

8. ETV6-RUNX1 fusion has been elucidated in the existing literature to be tightly linked to leukemia, but the conclusions of the present study would be meaningless if it just proved the results of the existing studies, and thus it is suggested that the beginning of the Discussion section should be modified to highlight the innovative conclusions of the present study.
9. It is recommended that a relevant description of the limitations of this paper be added, and that it be clarified what research ideas will be followed up to overcome some of the limitations and shortcomings that appear in this paper, especially a general description of the cellular or tissue experiments that will be followed up.
10. The Conclusion section of this study does not systematically summarize the main results of this paper, and it is recommended that this be modified to highlight the role of ALL-associated cell subsets in the regulation of CNV, the immune microenvironment, and to systematically elucidate the main findings of this study.

---

## Round 0.2 · accepted · Accept

Both reviewers have found your revisions satisfactory and have recommended acceptance. Your thorough response to the previous comments and the improvements made to the manuscript are appreciated

Reviewer 1 ·

Basic reporting

Thank you for the invitation from the editor again. In this study, the author elucidated the specific B progenitor cell clusters that affect the progression of B-cell acute lymphoblastic leukemia and their potential molecular mechanisms in the disease. The overall experimental design and analysis logic are rigorous and smooth, and the results and conclusions are mutually supportive. The revised manuscript has improved many details. They carefully responded to the reviewer's questions, and I have no further comments.

Experimental design

no comment

Validity of the findings

no comment

Reviewer 2 ·

Basic reporting

no comment

Experimental design

no comment

Validity of the findings

no comment

Additional comments

The aim of this study was to reveal the subset of B progenitor cells that drive the cell cycle in ETV6-RUNX1 positive B cell acute lymphoblastic leukemia (ALL) and its molecular mechanisms. The study first processed ALL related data obtained from public databases through single-cell clustering and revealed the basic characteristics of chromosome-related mutations in ALL samples through mutation analysis. Follow-up studies have focused on the effects of interactions between cell subsets on ALL progression. In addition to bioinformatics analysis, the study included numerous cell experiments to verify the effects of ALL progression-related target genes on the migration and invasion capabilities of cell line HAP1. In short, this study is a relatively comprehensive study, and after revision, the manuscript meets Peerj's publication requirements.